# A Critical Gap in Seagrass Protection: Impact of Anthropogenic Off-Shore Nutrient Discharges on Deep *Posidonia oceanica* Meadows

**DOI:** 10.3390/plants12030457

**Published:** 2023-01-19

**Authors:** Judit Jiménez-Casero, Maria Dolores Belando, Jaime Bernardeau-Esteller, Lazaro Marín-Guirao, Rocio García-Muñoz, José Luis Sánchez-Lizaso, Juan Manuel Ruiz

**Affiliations:** 1Department of Marine Sciences and Applied Biology, University of Alicante, 03690 San Vicente del Raspeig, Spain; 2Seagrass Ecology Group (GEAM), IEO, CSIC, Centro Oceanográfico de Murcia, 30740 San Pedro del Pinatar, Spain

**Keywords:** deep *Posidonia oceanica* habitats, off-shore anthropogenic pressures, nutrient loads, urban waste water, fish farm

## Abstract

In the Mediterranean, anthropogenic pressures (specifically those involving nutrient loads) have been progressively moved to deeper off-shore areas to meet current policies dealing with the protection of marine biodiversity (e.g., European Directives). However, conservation efforts devoted to protecting *Posidonia oceanica* and other vulnerable marine habitats against anthropogenic pressures have dedicated very little attention to the deepest areas of these habitats. We studied the remote influence of off-shore nutrient discharge on the physiology and structure of deep *P. oceanica* meadows located nearest to an urban sewage outfall (WW; 1 km) and an aquaculture facility (FF; 2.5 km). Light reduction and elevated external nutrient availability (as indicated by high δ^15^N, total N and P content and N uptake rates of seagrass tissues) were consistent with physiological responses to light and nutrient stress. This was particularly evident in the sites located up to 2.5 km from the WW source, where carbon budget imbalances and structural alterations were more evident. These results provide evidence that anthropogenic nutrient inputs can surpass critical thresholds for the species, even in off-shore waters at distances within the km scale. Therefore, the critical distances between this priority habitat and nutrient discharge points have been underestimated and should be corrected to achieve a good conservation status.

## 1. Introduction

As a consequence of the overload of human activities on the coast, the external nutrient loads on the marine environment (e.g., delivered from urban and industrial effluents, agriculture and aquaculture) have dramatically increased in recent decades [1,2,3]. The resulting excess of nutrients and organic matter has led to eutrophication processes in many coastal marine systems worldwide, which involves the decline of vulnerable and ecologically relevant marine habitats, such as corals and seagrasses [4,5,6]. Seagrasses are widespread engineering species that form complex tri-dimensional habitats (meadows) that sustain some of the most productive ecosystems on Earth and provide valuable ecological and socioeconomic services [7]. Seagrass meadows, however, are globally threatened by multiple anthropogenic pressures [8,9] and have experienced an accelerated decline in recent decades [10].

The endemic seagrass *Posidonia oceanica* is the most conspicuous and widely distributed in the infralittoral bottoms of the Mediterranean Sea, which forms almost continuous and extensive meadows from very shallow environments (ca. > 0.5 m) up to a maximum depth of 20–40 m [11]. This species is highly sensitive to human-induced environmental disturbances, particularly those involving changes in water and sediment quality (i.e., turbidity, nutrient and/or sediment organic enrichment and anoxia), which are characteristic of eutrophicated coastal areas [12,13,14,15]. Eutrophication is considered a major driver of local seagrass degradation and loss worldwide [16,17]. Although many studies have been focused on the analysis of seagrass responses to the environmental impact of nutrient over-enrichment, e.g., [11,14,18], our understanding of the relationships between anthropogenic pressures and seagrass ecological status is still very limited [19,20,21], and most of the available evidence has historically been biased towards more accessible shallow areas [4,22,23,24,25]. In comparison, *P. oceanica* meadows close to its depth distributional limits remain understudied and very little is known about their relationships with anthropogenic nutrient inputs, despite being considered highly vulnerable to environmental changes (overall those affecting light availability) [26,27].

In the legal framework, the implementation of the European directives (The Habitat Directive 92/43/EEC (HD) and the Water Framework Directive 2000/60/EC (WFD)) prohibited anthropogenic discharges in marine areas occupied by *P. oceanica* to ensure the protection and conservation of this and other priority habitats and to improve the ecological condition of marine water bodies and ecosystems in EU countries. This implies a diversion of anthropogenic pressure towards deeper, off-shore waters, away from the distributional limits of these and other vulnerable habitats [28]. However, the more recently implemented Marine Strategies Framework Directive 2008/56/EC (MSFD) has just identified deep *P. oceanica* meadows as one of the major gaps in biodiversity conservation and management of Mediterranean coastal zones, even in those countries who commonly comply with the implementation of previous HD and WFD Directives, such as Spain, Italy and France [29,30]. Several explanations account for this inadequate, undesirable situation. In the context of the WFD, the focus has been on *P. oceanica* meadows in shallow and intermediate depths (i.e., up to a maximum depth of ca. 15 m). In many protected areas included in the 2000 Nature Network, accurate and complete *P. oceanica* distribution maps are not yet available, especially in the deepest areas, which hinder the implementation of effective management strategies [31]. In addition, few marine protected areas have approved conservation and management plans that are effectively operational, i.e., with adequate and efficient monitoring programmes. This situation has resulted in the limited availability of information on the status of the deepest *P. oceanica* habitats, which prevents their ecological assessment by EU Directives.

In this context, a critical issue for the marine spatial planning of human activities is the minimum distance at which anthropogenic pressures must be moved away from the deep limits of *P. oceanica* to avoid degradation and habitat loss, e.g., [32]. However, this information is not generally available for this and other vulnerable habitats as it represents a difficult scientific challenge. For the establishment of these distances, it should be firstly considered that *P. oceanica* can respond to increasing pressures in a non-linear way [33], as evidenced for other aquatic systems, e.g., [34]. Seagrass species have mechanisms to cope with certain levels of, for example, light limitation or nutrient stress [13,35], but once a threshold is reached, the habitat may sharply precipitate to a degraded alternative state [36,37,38,39]. Secondly, once the habitat collapses, the recovery will be extremely slow (decades) or even impossible in this slow-growth species [15,40]. These circumstances should obligate coastal managers to take extreme precautions when deciding the distance between anthropogenic discharges and the nearest deep meadow habitats, overall if critical thresholds are unknown. Despite this lack of basic knowledge, it is widely assumed by scientists and coastal managers that the spatial extent of the influence of nutrient sources (e.g., aquaculture effluents) is localised (hundreds of metres), and, therefore, ‘safety distances’ of 500 m are considered sufficient to avoid their impact on neighbouring seagrass meadows [14,32,41,42,43]. However, the spatial extent of these pressures can vary widely depending on many factors related to species-specific traits, local conditions (currents fields, depth, etc.) and effluent characteristics [23,29,44,45]. In fact, some studies have shown that nutrient discharges from off-shore sources can reach the depth limits of *P. oceanica* meadows at distances of more than 1 km [29,33]. This suggests that: (i) the critical distances between *P. oceanica* meadows limits and anthropogenic nutrient sources could have been underestimated by scientists and coastal managers and (ii) the conservation status of deep *P. oceanica* meadows could have been altered or threatened by these diffused anthropogenic pressures even in coastal zones, where such pressures were moved away towards deeper off-shore marine areas.

From the situation described above, there is an urgent need to address the relationships between the ecological status of seagrasses and the influence of off-shore anthropogenic nutrient inputs in *P. oceanica* depth limits. Accordingly, the main goal of this study was to investigate the remote influence of off-shore anthropogenic nutrient sources on the physiology and structure of deep *P. oceanica* meadows. Therefore, we analysed the relationships between environmental conditions and *P. oceanica* traits at four meadow sites (26–27 m) at different distances from two anthropogenic nutrient sources: an urban sewage outfall and a fish farm complex. In situ measurements of light availability and sediment organic matter were carried out to characterise the effect of anthropogenic pressures on environmental conditions at each site, which was further assessed by quantifying nitrogen-stable isotopes (δ^15^N) and trace metals (Copper (Cu) and Nickel (Ni)) in plant tissues as these are typical tracers for such anthropogenic discharges [46,47,48,49]. At each site, seagrass variables representative of key physiological processes (i.e., photosynthesis and nitrogen metabolism) and meadow structure were also measured as they are proven to be suitable descriptors of seagrass ecosystem status and performance under eutrophic conditions [29,41,50,51]. The results obtained in this study have been discussed to understand the capacity of deep *P. oceanica* meadows to withstand diffusive nutrient inputs but also to get some insight into the determination of the spatial scale of ‘safety distances’ that could help coastal managers to preserve deep *P. oceanica* meadows from these kind of off-shore human activities in the long term.

## 2. Results

### 2.1. Univariate Analyses

#### 2.1.1. Variations in Abiotic Factors

Continuous light recordings revealed that, both in summer and autumn, the M1, M2 and M3 (i.e., up to 2.5 km from discharge sources) irradiance levels (i.e., daily PAR irradiance curves (E, Appendix A) and daily photosynthetic photon flux (DPPF, Table 1)) showed reduced values relative to M4, although this light reduction was less pronounced in M3 than in sites closer to WW (M1 and M2; Table 1). Accordingly, both M1 (summer and autumn) and M2 (autumn) received irradiance levels below 11% of subsurface irradiance (E_0_), while in M3 and M4, the light availability was clearly above in both seasons (Table 1). The sediment organic matter content (% OM) from the sites located close to the WW or the FF was significantly higher than in the distant site (M1 = M2 > M3 > M4, Table 1).

#### 2.1.2. Variations of Biological Indicators of Anthropogenic Pressure

Both the rhizomes and epiphytes of *P. oceanica* showed a progressive decrease in δ^15^N from M1 to M4 (Figure 1). The isotopic N signal was significantly higher in plants from M1, M2 and M3 (average of 4.59‰) than in the reference site M4 (average of 3.87‰), which was located within the range reported for unaltered deep *P. oceanica* meadows in the same study area [29]. A higher nutrient content (% N and % P) was found in plants closest to the WW source (M1 and M2; Figure 1), which also displayed the lowest C/N and C/P ratios (Appendix A). The Ni concentration in the rhizomes of the M1 plants was up to 1.6-fold higher than in plants from the other three sites (Appendix A). Cu content in rhizomes in M3 showed the lowest values compared to the other sites (Appendix A).

Ammonium uptake kinetics in *P. oceanica* was fitted to a Michaelis–Menten model (R^2^ > 0.98), where uptake rates from M1–M3 did not reach saturation, while M4, with a lower environmental nutrient availability, did (Appendix A). Ammonium uptake descriptors in *P. oceanica* leaves decreased progressively with increasing distance from the nutrient sources, (i.e., from M1 to M4; Figure 1f–h). Maximum uptake rates (V_max_) were 4.2, 2.6 and 2 times higher in plants from M1, M2 and M3 than reference ones (M4), while half-saturation constants (K_m_) were 8.5, 4.6 and 3.3 times higher, respectively. In contrast, the ammonium uptake efficiency (α_uptake_) in plants from M4 was 31% higher, on average, relative to plants located close to off-shore nutrient sources.

#### 2.1.3. Physiological Responses of *P. oceanica* Plants

Photosynthetic pigments (Chlorophyll *a*, *b* and carotenoids) showed similar patterns of variation in both seasons (Figure 2a,b). In summer, plants located closer to the WW discharge point (M1 and M2) had higher pigment concentrations than those from M3 and M4. In autumn, only plants located at 1 km to the WW showed significant increases regarding all the other plants (M2, M3 and M4).

The photosynthetic responses derived from P–E curves reflected a significant reduction in gross photosynthesis (gross-P) and respiration (R) rates in plants nearest to the WW and FF sources (M1, M2 and M3) in both seasons, with respect to plants from the reference site (M4; Figure 2). This pattern was also observed for saturation irradiance (Ek) in autumn and compensation irradiance (Ec) in summer, with the M1 site showing the lowest Ec values in autumn (Appendix A). Plants located 2.5 km from the sources (M2 and M3) showed a lower photosynthetic efficiency (α) value in summer, as well as M2 in autumn (Appendix A). The relative electron transport rate (r-ETR) was only significantly reduced in plants located 1 km from WW (M1) in autumn (Appendix A). The maximum quantum yield (Fv/Fm) was significantly higher in M1 and M2 than the most distant site (M4) in summer (Appendix A). In autumn, M1 showed the highest Fv/Fm values.

The daily metabolic carbon balance (CB) and the daily periods of saturation and compensation (Hk and Hc) were higher in summer than in autumn (Figure 3). Differences in CB among the studied plants were only observed in autumn, with the lowest mean values (even negative) for those closest to WW. At the M1 meadow site (1 km from WW), Hk was also the lowest in autumn, but it was 1.33 h longer at stations M2 and M3 (2.5 km from the nearest source), relative to M4. The Hc values did not vary among sites, showing mean values (±SE) of 13.77 ± 1.26 h in summer and 9.78 ± 0.89 h in autumn.

The non-structural carbohydrate content (starch and soluble sugars) was higher in summer than in autumn (Figure 4). The soluble fraction displayed significant variations among the studied sites but not the starch fraction (Figure 4). Plants from M1 and M2 showed a lower soluble carbohydrate content than plants from the other sites (M3 and M4) in summer. In autumn, the M2 plants also showed the lowest content.

#### 2.1.4. Changes at the Structural Meadow Level

All structural meadow descriptors showed significant differences among the studied sites (Figure 5). Mean values of meadow cover, shoot density and vertical rhizome length in meadows closest to the WW source (M1 and M2) were significantly lower than in the other two meadow sites (M3 and M4). Similarly, no differences in structural variables were found between M3 and M4 in previous studies 18 years before [29]. However, in this study the meadow cover at M3 showed a higher variability, with intermediate mean values higher than those closest to the WW effluent but still lower than M4. The highest proportion of plagiotropic and shortest rhizomes was found in the area nearest to the WW source (M1 and M2). No statistical differences in other measured seagrass descriptors (shoot size (37.5 ± 1.4 cm^2^ shoot^−1^), epiphyte load (0.8 ± 0.05 mg cm^−2^) and overgrazing (6.13 ± 1.6%)) were observed between meadow sites.

#### 2.1.5. Synthesis of Variable Responses

In order to facilitate the understanding and interpretation of the variables responses, as well as their relationships with anthropogenic pressures, a synthesis of the main effects reported for each variable in the previous sections is provided in Table 2. In summary, the most significant departures from the reference condition (i.e., M4) were observed at the M1 meadow site, followed by M2, both in environmental variables (abiotic factors) and in all seagrass indicators (biotic factors). M3 also showed a certain degree of alteration but to a much lesser extent than that reported for M1 and M2. The most affected variables in M1 and M2 were those related to the nutrient content, δ^15^N (overall in epiphytes) and nitrogen uptake, and variables related to the photosynthetic carbon metabolism and meadow structure (meadow cover and shoot density).

### 2.2. Multivariate Analysis

The principal component analysis (PCA) conducted with the bioindicators of human pressure showed a distribution of the four meadow sites along axis I, reflecting the highest signal of external nutrient inputs on the M1 site and the lowest on M4 (M1 > M2 > M3 > M4; Figure 6). This axis represented 85% of the explained variance and was positively correlated (r > 0.70) with C/N and C/P ratios and α_uptake_, but negatively correlated with δ^15^N, % N, V_max_, k_m_ and Fv/Fm (Appendix A). The second ordination axis accounted for only 11% of the total variance and was highly correlated with the Cu content in rhizomes and the r-ETR (Appendix A).

The descriptors of light availability (PAR irradiance, DPPF) and sediment (OM) quality and the distance to the nearest pressure source also showed a high correlation with axis I of the PCA (Figure 6, Appendix A), reflecting a decrease in OM content and higher DPPF with the distance (from M1 to M4) to any source of anthropogenic inputs. The high correlation of both the bio-indicators and the abiotic factors with axis I reflected similar variations and different degrees of nutrient loads and environmental disturbance among sites (M1 > M2 > M3 > M4). Based on these high correlations we could generate a new standardised variable (0–1) from the score values of axis I that can be used in this study as an index of the degree of pressure influence (DPI). The maximum value corresponded to the meadow site M1 and the minimum to the reference site M4; the other two meadow sites occupied intermediate positions, with DPI values higher in M2 (0.63) than in M3 (0.39).

## 3. Discussion

Results obtained in this study provide robust evidence of physiological and structural alterations of *P. oceanica* related to the remote influence of off-shore anthropogenic inputs on meadow depth limit, even at distances up to 2.5 km. Plant tissues (rhizomes) and epiphytes were ^15^N-enriched at M1–M3 sites, relative to plants from the reference site M4, which were within natural ranges previously reported for this variable in this and other study areas, e.g., [29,51]. This isotopic enrichment is a clear indicator of the remote influence of nitrogen inputs of anthropogenic origin as it has been reported for seagrasses in relation to the influence of fish farm loadings [51,52] and sewage wastes [53,54] and in general for other macrophytes from eutrophicated zones [46,55]. Changes in abiotic factors described for meadow sites closest to FF and WW discharges (i.e., reduction in light availability and sediment organic enrichment) were also consistent with typical environmental deterioration associated with coastal eutrophication [56,57], and showed a strong correlation with bio-indicators of external nutrient availability (i.e., δ^15^N, N and P content and nutrient uptake kinetics; Figure 1). The close relationship of these variables with the first PCA axis allows us to develop the DPI index (degree of pressure influence), that integrates the information provided by each variable into a single value that reflect the relative level of pressure influence on each meadow site. This DPI index was closely and negatively correlated with the distance to anthropogenic sources, as represented in Figure 7. The maximum DPI value corresponded to meadow site M1, the closest to the anthropogenic discharge (i.e., 1 km from the WW in this case), where the highest nutrient (N and P) and metal (Ni) contents were observed. The meadow sites M2 and M3, which were located 2.5 km away from their respective nearest nutrient source (i.e., WW and FF, respectively), showed intermediate DPI values. Both sites also showed elevated an N isotopic signal of plant tissues, but this influence was higher in M2 plants (as was also reflected in P and C/P ratio in rhizomes and C/N in epiphytes; see Table 2). This suggests a stronger influence of remote WW discharge on the studied deep *P. oceanica* meadow than that closest to the FF effluents.

In eutrophicated environments, the reduction in light availability for benthic macrophytes is mainly attributed to an increased seawater turbidity [17,50,58,59]. Under such conditions, light limitation on seagrasses can also be indirectly enhanced through known nutrient-induced interactions, such as epiphyte overloading and/or overgrazing, e.g., [22,41,60,61]. However, no evidence of the operation of these last mechanisms was observed in deep *P. oceanica* meadows in this study, which contrasts with their importance reported in shallow *P. oceanica* meadows, e.g., [22]. Independently of the mechanism involved, our results suggest that such light reduction associated with the anthropogenic influence could have contributed to enhanced light limited conditions already existing in seagrass depth limits and affected seagrass performance and integrity. At the reference site (M4), light availability was very close to levels predicted for *P. oceanica* at its depth limit (10–16% E_0_), according to the experimental determinations of minimum light requirements (MLR) for growth obtained for this species in previous studies [62,63]. In fact, the structural characteristics of this meadow (shoot density and meadow cover) were similar to those expected for the species at its depth limit in this Mediterranean region [64]. In contrast, light availability at the influenced meadow sites was consistently below such levels, which could have induced further stress on seagrass physiology and growth, particularly at those sites closest to the urban sewage effluent WW, where structural alterations were rather evident. As explained below, photo-physiological responses reported in this study support this hypothesis.

The main photo-physiological adjustments consisted of a decrease in both net photosynthesis, r-ETR and respiration rates, which are the most common responses of *P. oceanica* and other marine macrophytes under light-limited conditions [26,27,62,65,66]. In shallow *P. oceanica* meadows, these photo-acclimative responses have been reported under more severe levels of light reductions [62,65], supporting the hypothesis that the deepest *P. oceanica* meadows are more sensitive to even small and moderate events of light reduction [26]. Plants from sites subjected to the most severe light reduction (i.e., M1 and M2) even showed a further photo-acclimative effort, as indicated by the increase in pigment content. This is a typical photo-acclimative response of this seagrass species to optimise light capture by the photosynthetic apparatus, as indicated by the increase in Fv/Fm [67]. In addition, higher nitrogen availability in these meadow sites could also be supporting this increase in photosynthetic pigments [68]. Overall, these photosynthetic responses enable the seagrass to adjust light requirements for growth to the reduced light availability, as denoted by the decrease in compensation and saturation irradiances (Ec and Ek, respectively), and to maintain its photosynthetic and productive capacity, as indicated by the maintenance of Hk (i.e., the daily period at which seagrass photosynthesises at saturating irradiance) [24,69]. These photo-acclimative mechanisms seem to only be effective in plants with the lowest anthropogenic influence (i.e., M3, close to the FF source), which had the capacity to maintain a carbon balance (CB) similar to that of the reference meadow, M4. By contrast, plants under the influence of the WW source (i.e., M1 and M2 with the highest DPI values) were unable to maintain CB. The influence of other pollutants, such as metals (as suggested by highest values in plants from the M1 site), could also help to explain some of these responses at the physiological level, since it could induce inhibitory effects on photosynthesis and carbon metabolism [70,71].

The reported imbalances of the photosynthetic carbon metabolism are consistent with the severe alteration of meadow structure (reduced shoot density and meadow cover) at the M1 and M2 sites, which, in turn, is in agreement with declining trends reported for these variables in the study area in the recent decades, e.g., [72]. *Posidonia oceanica* and other slow-growing seagrasses form perennial structures with relatively low ratios of aboveground to belowground biomass, implying high metabolic costs for their maintenance [73,74]. In these species, biomass reduction is an effective acclimative mechanism to compensate for metabolic imbalances caused by light reduction and other environmental stressors, as it is observed across the natural gradients of light reduction (e.g., with depth) [26,63,75]. Changes in rhizome morphology reported in these altered meadows (i.e., the dominance of plagiothropic rhizomes and shorter vertical shoots) could also be interpreted in this context [76], since it is also a consequence of an over-simplification of standing non-photosynthetic structures. Therefore, the simplified meadow structure reported in sites M1 and M2 could merely reflect the acclimation of deep *P. oceanica* meadows to the reduced light availability caused by the influence of anthropogenic pressures.

In highly disturbed environments, multiple stressors can operate to induce seagrass degradation [12,32]. In our case, in addition to light limitation, other abiotic factors, such as nutrient excess, sediment anoxification or metals, could help to explain the reported metabolic and structural alterations of deep meadow sites. Under these scenarios, complex interactions between light, nutrient excess and the organic enrichment of sediments are known potential drivers of seagrass decline, although it would also depend on other local factors and species traits [57,77,78]. In over-enriched environments, *P. oceanica* and other seagrass species are unable to down-regulate nitrogen uptake [78], which should be immediately assimilated into organic forms [13,79] to prevent intracellular toxicity effects [80,81]. This process involves large metabolic costs and a carbon drain that is provided by photosynthesis and/or by the mobilisation of the non-structural carbohydrate pool. The reported reduction in carbohydrate content of plants from M1 and M2 sites could be related with these N-induced metabolic alterations and could jeopardise seagrass growth and biomass, particularly during periods of light limitation (e.g., winter). In addition, the potential negative effect of OM-enriched sediments could contribute to the reported physiological and structural alterations. Although the OM content we found at M1 and M2 (3–4%) was not as high as that observed in other nutrient-impacted ecosystems (9–25%; [12,15]), its impact on the seagrass status could be potentially exacerbated at sites with reduced light availability, e.g., [77], although this interaction has not been evaluated in deep *P. oceanica* meadows.

Meadow sites M2 and M3 differed in DPI values, despite being located at the same distance from the nearest nutrient source (2.5 km). This can easily be explained both by the high spatial variability of local factors (e.g., hydrodynamic exposure, current fields, wind regime, etc.) and the variability in the characteristics of each anthropogenic nutrient source (e.g., frequency, duration and intensity of discharges, composition, etc.) [43,82]. However, this differential influence level was not significant enough to explain the severe damages reported in the *P. oceanica* meadow at the M2 site. A possible explanation could be the existence of pressure thresholds (or tipping points), beyond which the habitat suddenly collapses and declines (Figure 7). As evidenced in terrestrial and marine ecosystems, even in seagrasses [33,83], these kinds of abrupt responses can be related to the existence of non-linear relationships between pressure influence and ecosystem state. These non-linear response models involve the existence of resilience mechanisms that are able to hamper the increase in pressure influence over time, as evidenced for some seagrass species, e.g., photo-acclimation, [84]. In fact, the maintenance of the habitat integrity in M3 could be explained by such mechanisms continuing to operate at that level of pressure influence. Based on these models, this could also mean that the seagrass habitat is closer to reaching its critical pressure threshold and that it could precipitate its shift to a degraded state in response to even a small increment of pressure from the nearest anthropogenic source (i.e., the FF) in the following years. The high variability of meadow cover observed in M3 is consistent with this possibility, as described in other aquatic ecosystems just before reaching the tipping point [34]. This hypothesis should be supported by additional data and study cases, although it is in line with other findings about this and other seagrass species through other methodological approaches, e.g., [33,84]. The adoption of this kind of non-linear models should be seriously considered by coastal managers in order to guarantee the preservation of *P. oceanica* meadows, since the recovery of these kinds of vulnerable habitats does not follow a linear pathway once they have collapsed.

## 4. Materials and Methods

### 4.1. Study Site and Sampling Design

The present study was conducted in 2016 along the southeastern coast of Spain (Mediterranean, Murcia Region) with two main anthropogenic off-shore nutrient sources: a fish farm complex (FF) and an urban wastewater treatment plant (WW). The FF source had a total annual fish production of 9000 tonnes (data provided by the General Directorate of Livestock, Fisheries and Aquaculture of Murcia Region), and the WW source had a flow rate of 40,000 m^3^/day and 26,000 m^3^/day in summer and autumn, respectively [85].

In this coastal area, *P. oceanica* forms an extensive and continuous meadow up to a maximum depth of 33 m [86]; as shown in Figure 8, which is included in the 2000 Natura Network of the Habitat Directive. The seagrass habitat is highly homogeneous in structure, substrate type and hydrodynamic exposure, and has a good conservation status [86]; as shown in Figure 8. Water quality in the area is highly oligotrophic, with surface currents flowing mainly parallel to the coastline in the NW and SW directions, with an average speed of 10 cm s^−1^ [29]; as shown Sánchez-Lizaso unpublished data. The dominant deeper currents (28 m depth) are in the SW and NE directions, and the average speed is 8–9 cm s^−1^ (CARM, https://caamext.carm.es/siom/index.php, accessed on 7 December 2021).

The seagrass status was assessed in four selected sites (M1–M4) located at the deep meadow margin (26–27 m) at different distances from anthropogenic discharges (Figure 8). Sites M1 and M2 were located, respectively, at 1 and 2.5 km from the off-shore WW discharge point (the closest nutrient source). Site M3 was located 2.5 km from the FF complex (and 4.5 km from the WW discharge point). Site M4 was placed 14.5 and 12.5 km from WW and FF, respectively, and was considered far enough away to be out of the influence of these and other pressures. This meadow site has been considered a reference site in previous studies, e.g., [35,87], and is within the marine protected area of Isla Grosa. Therefore, this sampling site (M4) will be hereafter referred to as a ‘reference’ site. The position of sampling stations M1–M3 were taken from previous studies performed in the same area so that some background information was already available [29,72].

At each meadow site, six linear transects of 10 m length were deployed perpendicularly to the depth meadow edge. Transects were placed by scuba divers every 15 m throughout the seagrass limit, leading to a total sampling area of 90 m^2^, a surface that can be considered representative for the characterisation of this habitat. For each transect, replicated measures and samples of each of the selected seagrass variables were obtained (see below). Seagrass sampling was carried out in summer (July) and autumn (October), when the *P. oceanica* was within its optimum of the annual production cycle, e.g., [73]. At each site, light availability was measured by installing two quantum sensors, and three sediment samples were also taken for their analysis (see below). After sampling, biological and sediment samples were transported to the laboratory for further analysis.

### 4.2. Abiotic Factors

Light availability (measured as PAR irradiance) and sediment organic matter were the environmental factors selected for the characterisation of the water column and sediment quality. These are key abiotic factors that are usually altered under the influence of anthropogenic nutrient discharges, to which seagrass health and status have been demonstrated to be particularly sensitive [15,41,50].

Light regimes were characterised at each sampling site and season, and the percentage of organic matter in the sediments was only measured1 in autumn. For the characterisation of light regimes, underwater PAR irradiance was measured 10 cm above the bottom using two spherical quantum sensors (Alec MDS MK5) that were separately installed on un-vegetated patches within the different meadow sites. At each site and season, sensors were programmed to record instantaneous measurements of PAR irradiance (as photosynthetic photon flux density, PPF, µmol quanta m^−2^ s^−1^) every 10 min for a period of 30 days. The daily photosynthetic photon flux density (DPPF, mol quanta m^−2^ d^−1^) was obtained for each day from the integration of daily PPF curves. PPF values at noon were also relativised to average PPF values and measured just underneath the sea level (E_0_) to estimate the percentage of the subsurface irradiance that reached the sea bottom (% E_0_). Estimates for this variable were only performed for calm sunny days of summer and autumn in order to obtain standardised values. Subsurface measurements of PPF were obtained using a cosine-corrected quantum sensor (LI-190SA; LI-COR) so that an inter-calibration between both types of sensors had to be performed in the laboratory. Three sediment samples were also taken at the extremes and centre of the sampling area, with cores (7 cm diameter; 15 cm deep) at each site. The organic matter content (OM %) was determined as the percentage of weight loss upon calcining dry sediment in a muffle furnace at 550 °C for 5 h [88].

### 4.3. Seagrass Bioindicators of Anthropogenic Pressure

Nutrient content (N, P), isotopic nitrogen (δ^15^N) and metal content (Ni and Cu) were measured in seagrass tissues (epiphytes and/or rhizomes), instead of in the seawater or sediments, since macrophytes are widely recognised as more reliable indicators of the external availability of nutrients and other pollutants in marine environments. Additionally, these variables are excellent physiological indicators of the metabolic stress caused by the anthropogenic influence [42,46,51,54,61,89].

For each site, 2 vertical shoots were randomly collected within each of the six transects (see above). In the laboratory, rhizomes and epiphytes tissues were separated from each shoot, which are considered to be the most robust compartments to detect changes associated with external nutrient inputs [51]. For each transect, rhizome and epiphytes tissues of the two shoots were pooled, and 2 mg of dry and ground samples of both tissues (*n* = 6) were encapsulated for determination of (i) N and C content (% N and % C) using a Carlo-Erba CNH elemental auto-analyser, and (ii) δ^15^N (‰) using an EA-IRMS (Thermo Finnigan). The concentrations of metal and phosphorous (% P) were determined for plants from each transect (*n* = 6) by acid digestion of 0.2 g of dry and ground samples of rhizomes using a microwave system (Ultrawave, Milestone Ethos Sel, Italy). The concentrations of Ni, Cu and P were determined using IC-OES (7600 Duo deThermo, Germany).

The epiphyte biomass separated from each shoot was standardised to the total surface area of that shoot to obtain epiphyte load (mg cm^−2^). The number of leaves per shoot bitten by macro-herbivores was also annotated to estimate herbivore pressure (as a percentage), a proxy of herbivore activity [22].

Uptake kinetics for NH_4_^+^ were determined by exposing *P. oceanica* leaves to ^15^N-labelled tracers in transparent split chambers, following methods described by Sandoval-Gil et al. [90]. Incubation was performed in 500 mL volume glass chambers with filtered seawater and an initial ammonium concentration <1 µM, using two mature healthy leaves, non-epiphyted, from six randomly collected shoots in autumn at each site. Four different concentrations (5, 25, 50 and 100 µM) of labelled ammonium (^15^NH_4_Cl at% = 99, Cambridge Isotope Laboratories) was applied during one hour under environmental conditions similar to the field (Temperature: 20 °C; Irradiance: 70 µmol quanta m^−2^ s^−1^). Six incubations (*n* = 6) were carried out for each of the ammonium concentrations and for each of the four sites, with a total of 96 incubations. At the end of the exposure, leaves were rinsed with deionised water and isotopic determinations were carried out, as explained above for natural samples.

Ammonium uptake rates (V) were calculated, as described by Sandoval-Gil et al. [90], at each treatment. Uptake rates were plotted against the substrate concentration (S, μM), and the uptake kinetic parameters were calculated using the Michaelis–Menten model:V = (V_max_ × S)/(K_m_ + S)
where V_max_ is the maximum uptake rate (μmol N g^−1^ DW h^−1^) and Km is the half-saturation constant (i.e., value at which V = V_max_/2, μM). The efficiency of nitrogen uptake (α_uptake_) was also calculated as V_max_/K_m_.

### 4.4. P. oceanica Physiological Features

Photo-physiological characteristics of *P. oceanica* plants were analysed using two non-epiphyted leaves of two randomly collected shoots along each transect at each site and during each season. Rhizomes of these shoots were used for carbohydrate analysis. For each transect and physiological variable, averaged values of two measurements were used as independent replicates (i.e., *n* = 6 replicates per meadow site).

To measure photosynthetic pigment content (Chloroplyll *a, b* and carotenoids), approximately 2–3 cm of mature leaves from six plants was homogenized, using buffered acetone (80%) and maintained overnight at 4 °C to ensure complete tissue disaggregation. After centrifugation (1000× *g* at 10 min), the absorbance of the supernatants was measured at 470, 646, 663 and 725 nm to calculate the pigment concentration, following the methods described in Lichtenthaler and Wellburn [91] and expressed per fresh weight (µg g^−1^ FW).

Maximum photosynthetic and respiratory rates (μmol O^2^ g^−1^ FW h^−1^) were measured using approximately a 2 cm^2^ leaf segment per shoot selected for pigment analysis. An incubation chamber with a Clark-type O_2_ electrode (Hansatech, UK) under controlled temperature similar to the field (20 °C) was used following the methods described in Marín-Guirao et al. [92]. Incubation consisted of a 5 min dark exposure followed by a 7 min exposure to six increasing irradiances (10, 20, 40, 100, 300 and 650 μmol quanta m^−2^ s^−1^) and a final 20 min darkness exposure to determine dark respiration (R). Net photosynthetic rates were plotted against the light intensities (P–E curve), and photosynthetic parameters were calculated as follows: the maximum net photosynthetic rates (net-P_max_) were determined by averaging the maximum values above the saturating irradiance (Ek), and gross photosynthesis (gross-P_max_) was calculated as the sum of net-P_max_ and R. Ek was calculated as the ratio net-P_max_/α, where α (photosynthetic efficiency) was calculated as the slope of the regression line fitted to the initial linear part of the P–E curve, and the compensation irradiance (Ec) was the intercept on the *X*-axis.

Chlorophyll *a* fluorescence was calculated to obtain the relative electron transport rate (r-ETR) and the maximum quantum yield of photosystem II (PSII; Fv/Fm), as a proxy to evaluate the function of the photosynthetic apparatus. The Fv/Fm was measured in all night dark-adapted leaves. The middle section of two mature leaves from six different shoots (*n* = 6) was exposed to 2 h of illumination, and later, following the methods described in Marín-Guirao et al. [93], rapid light curves (RLCs) were generated using a Diving-PAM portable fluorometer (Walz, Germany). Each curve involved a 20 s exposure of 9 increasing irradiances (0, 2, 10, 38, 61, 128, 158, 295 and 406 μmol m^−2^ s^−1^), and the relative electron transport rate (r-ETR) was analysed using the PAM WinControl programme (Walz, Germany).

The mean daily compensation (Hc) and saturation (Hk) periods and daily metabolic carbon balance (CB) were selected as proxies of plant productivity [24,79,94]. The Hk and Hc periods were calculated by averaging the number of hours per day in which irradiance values exceeded the corresponding Ek and Ec values. The Ek and Ec were obtained from the P–E curves. The daily carbon balance was calculated according to the Michaelis–Menten function (P = [gross − P_max_ E/(E + Ek)] + R [95]), where P is net photosynthesis and E is the irradiance measured in the field, the other parameters were mentioned above. The field irradiance measurements (i.e., every 10 min) were entered into the function to generate estimates of net production, which were integrated across 24 h periods to yield daily net production (*n*= 30 days). The net production (mg C g FW^−1^) was calculated, as described in Marín-Guirao et al. [96], based on [97].

Carbohydrate measures were made from the 2–3 cm apical part of selected rhizomes following the method described by Marín-Guirao et al. [93]. Soluble sugars and starch were determined spectrophotometrically, using an anthrone assay and expressed as a percentage of dry weight (% DW).

### 4.5. Meadow Structure Descriptors

Five different structural metrics were measured by scuba divers along the six linear transects at each site in autumn. At the structural level, meadow status was characterised by measuring meadow cover, shoot density within meadow patches, shoot size (see above), rhizome height and the proportion of plagiotropic (colonising) rhizomes [98,99,100].

Meadow cover (as a percentage) was visually estimated in 1600 cm^2^ quadrats allocated every meter along each transect (i.e., 10 measurements per transect), as described in Ruiz et al. [29]. Shoot density (shoots m^−2^) was estimated by counting the number of shoots within two 400 cm^2^ quadrats randomly located along each linear transect. The vertical rhizome length (cm) was measured, using a ruler, on four orthotropic rhizomes that were randomly located along each transect. This measure corresponded to the distance from the base of each shoot (i.e., its insertion point in the plagiotropic rhizome) to the ligule of the outermost leaf. The percentage of plagiotropic rhizomes was calculated by the proportion of horizontal rhizome apexes relative to the total number of shoots counted in six of the 1600 cm^2^ squares. Overall, the measurements were averaged by transect (*n* = 6).

### 4.6. Statistical Analysis

The principal component analysis (PCA) was used to explore the potential relationship between spatial patterns of the studied meadow sites with anthropogenic nutrient sources (WW and FF). The unconstrained analysis was applied using normalised data of the seagrass bioindicators from the anthropogenic influence. Abiotic factors selected as indicators of water and sediment quality (DPPF and % OM) and the distance to the closest anthropogenic N loads were superposed to the obtained multivariate ordination. Based on methodologies used in the development of known indexes of *P. oceanica* status, e.g., [98,101,102], if the biological indicators and abiotic factors are highly correlated with any of the PCA axes, this axis could be extracted as a new variable (based on its score value). This new variable can be interpreted in terms of the degree of external pressure influence (DPI), which is derived from the studied anthropogenic pressures (WW and FF) and can be standardised between 0 and 1, with 1 being the maximum degree of pressure influence of anthropogenic discharges and 0 corresponding to an uninfluenced condition (i.e., natural or quasi-natural). The PCA was performed using CANOCO software version 5.0 (Microcomputer Power Ltd., Ithaca, NY, USA).

To test significant differences for each response variable of *P. oceanica* among the four studied sites, a one-way analysis of variance (ANOVA) and a post-hoc Tukey test were applied separately to the data from each season. Before carrying out ANOVA, the data were checked for the assumptions of normality and homoscedasticity and transformed when necessary. If homoscedasticity and normality of model residuals were not met, a more conservative approach was applied reducing the significance level (*p* ≤ 0.01) or using the post-hoc Bonferroni correction [103,104] in the R software (R Core Team, 2020).

## 5. Conclusions

Our findings strongly suggest that the conservation status of deep *P. oceanica* meadows can be altered by even remote, off-shore anthropogenic nutrient sources located at distances of up to 2.5 km away. Consequently, conservation actions implemented in the area for the protection of this priority habitat could have been offset by the effects of these diffuse pressures on the deepest seagrass meadows. This contrast with ‘safety distances’ of several hundred metres (500 m on average), generally assumed between *P. oceanica* depth limits and anthropogenic pressures [32,33,42,43,100,105]. This highlights an urgent need to review the criteria used by coastal managers to establish more realistic safety distances, which could have been underestimated in this and many other cases [106]. This issue is particularly relevant in the current scenario of global climate change, since warming can make *P. oceanica* more vulnerable to anthropogenic disturbances [107,108], and deep *P. oceanica* meadows have been demonstrated to be more vulnerable to warming than shallower meadows [87]. Once the pressure influence surpasses seagrass tolerance thresholds, the recovery of slow-growth species, such as *P. oceanica*, is unlikely even if the pressure ceased [109]. Therefore, coastal managers should apply safety distances keeping deep *P. oceanica* meadows well away from the critical thresholds of seagrass tolerance to anthropogenic pressures. However, the knowledge of these thresholds is absent in most cases, which should make the application of the precautionary principle mandatory [110]. In this context, our results support the need to integrate the use of physiological indicators into monitoring programmes of environmental impact studies, since they have been demonstrated to be a useful and efficient tool, in addition to having great potential to inform on pressure influence before reaching the tipping point, as has also been supported by previous studies [29,51,101,102].

## Figures and Tables

**Figure 1 plants-12-00457-f001:**
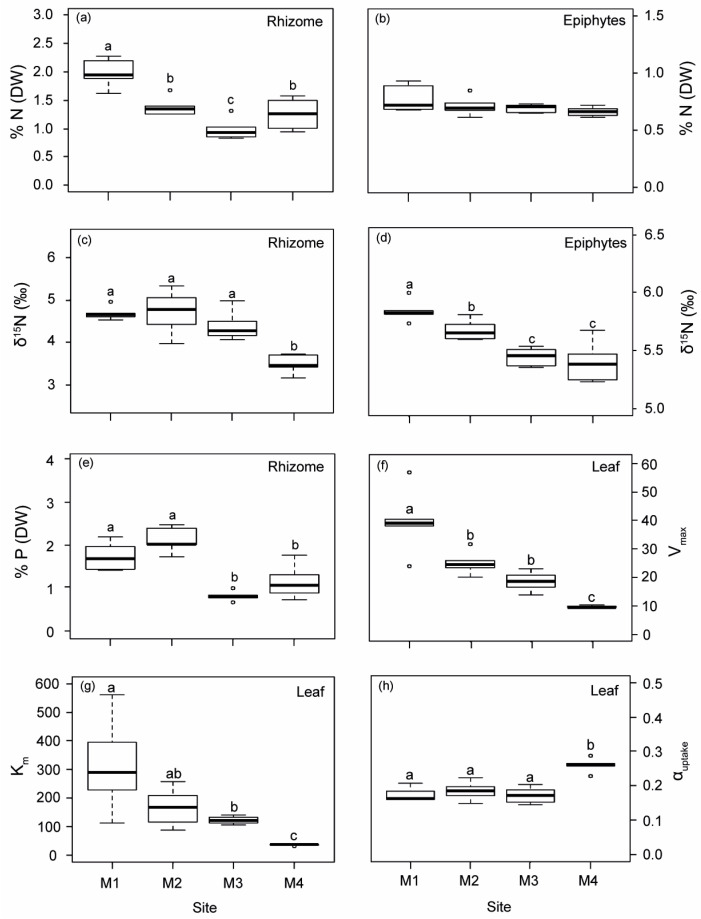
Biological indicators of nutrient loads (rhizome and epiphytes) and leaf ammonium uptake kinetic parameters measured in *P. oceanica* from the four sampling deep meadow sites (M1–M4) in autumn. (**a**,**b**) Nitrogen content (% N), (**c**,**d**) N isotopic signature (δ^15^N), (**e**) phosphorous content (% P), (**f**) maximum uptake rates (V_max_), (**g**) half-saturation constant (K_m_) and (**h**) uptake efficiency (α_uptake_). Different letters denote significant differences among sites (Tukey test, *p* < 0.05).

**Figure 2 plants-12-00457-f002:**
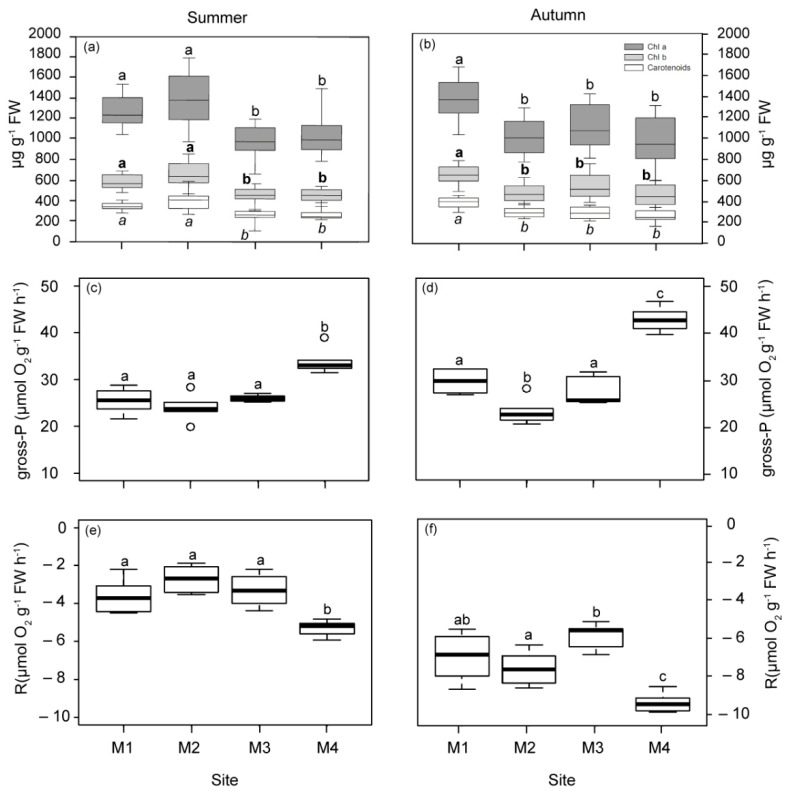
Pigment content and photosynthetic parameters measured in plants from the four studied deep meadow sites (M1–M4) in summer and autumn (left and right, respectively). Chlorophyll *a*, *b* and carotenoids (**a**,**b**), gross photosynthesis (gross-P; (**c**,**d**)), respiration rates (R; (**e**,**f**)). Different letters denote significant differences among sites (Tukey test, *p* < 0.05).

**Figure 3 plants-12-00457-f003:**
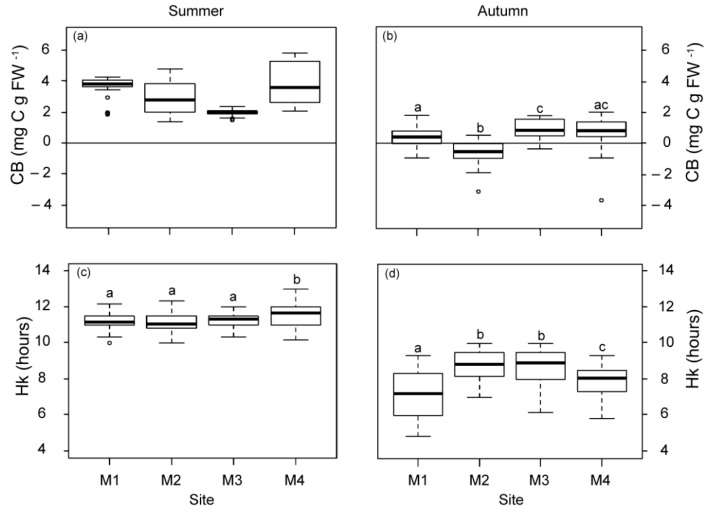
Daily metabolic carbon balance (CB; (**a**,**b**)) and daily saturation period (Hk; (**c**,**d**)) at the four meadow sites (M1–M4) in summer and autumn. Different letters denote significant differences among sites (Tukey test, *p* < 0.05) and the line indicates zero.

**Figure 4 plants-12-00457-f004:**
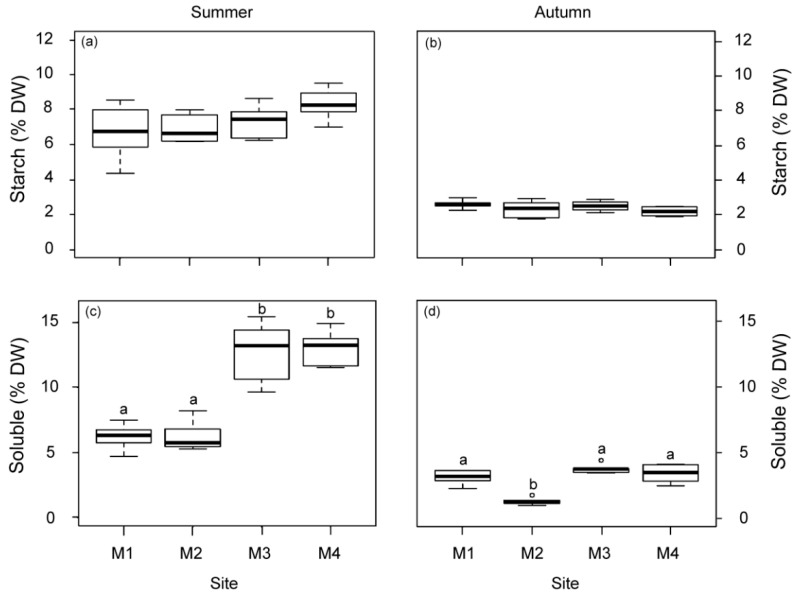
Non-structural carbohydrate content in rhizomes, starch (**a**,**b**) and soluble sugars (**c**,**d**) of *P. oceanica* plants from the four deep meadow sites (M1–M4) in summer and autumn. Different letters denote significant differences among sites (Tukey test, *p* < 0.05).

**Figure 5 plants-12-00457-f005:**
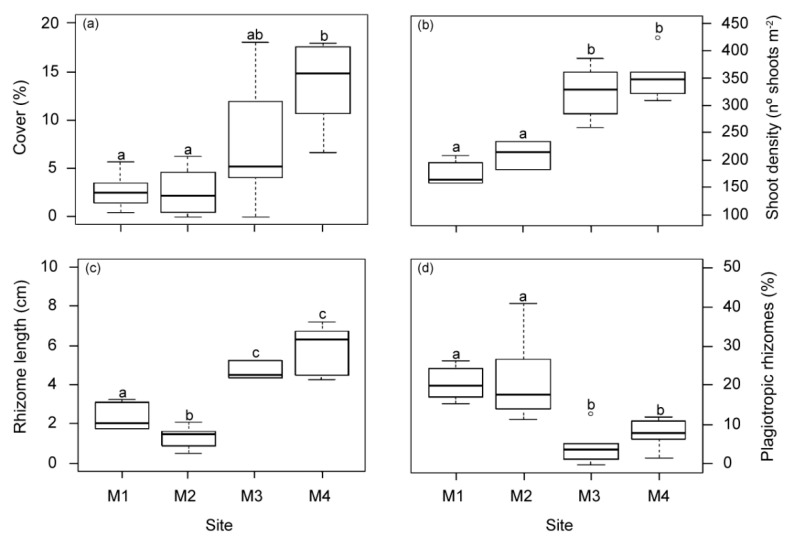
Meadow structural descriptors measured at the four meadow sites (M1–M4) in autumn. (**a**) Meadow cover, (**b**) shoot density, (**c**) rhizome length and (**d**) % plagiotropic rhizomes. Different letters indicate significant differences among sites (Tukey test, *p* < 0.05).

**Figure 6 plants-12-00457-f006:**
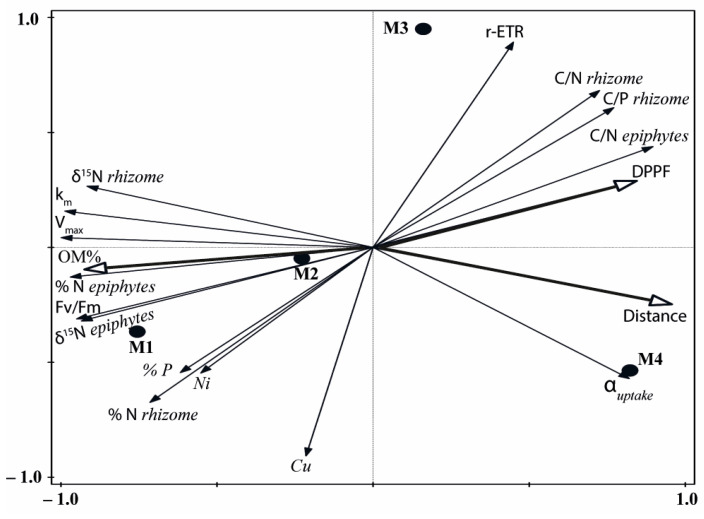
Unconstrained principal components analysis (PCA) of biological indicators of anthropogenic pressure from the four studied deep meadow sites (M1–M4). Abiotic factors (DPPF, OM % and distance) are superimposed on the plot.

**Figure 7 plants-12-00457-f007:**
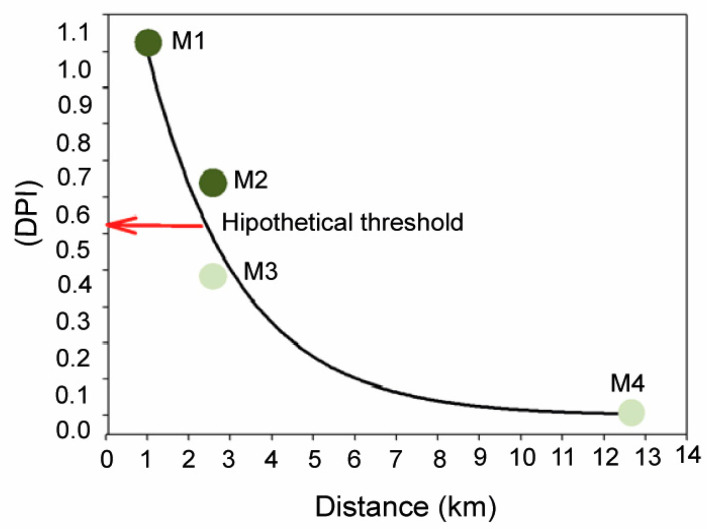
Relationship between the degree of pressure influence (DPI, score of PCA axis I) and the distance from each studied site (M1–M4) to the nearest anthropogenic nutrient source. The solid line corresponds to the theoretical exponential decay model fitted to the data to illustrate the inverse relationship between the two variables. As it is suggested in the text, the sudden shift from a healthy meadow state (i.e., with the integrity of the meadow structure intact; light green circles) to a degraded state (i.e., with an altered meadow structure; dark green circles) could be explained by the existence of a hypothetical threshold (red arrow) as a result of a non-lineal relationship between seagrass status and the pressure intensity.

**Figure 8 plants-12-00457-f008:**
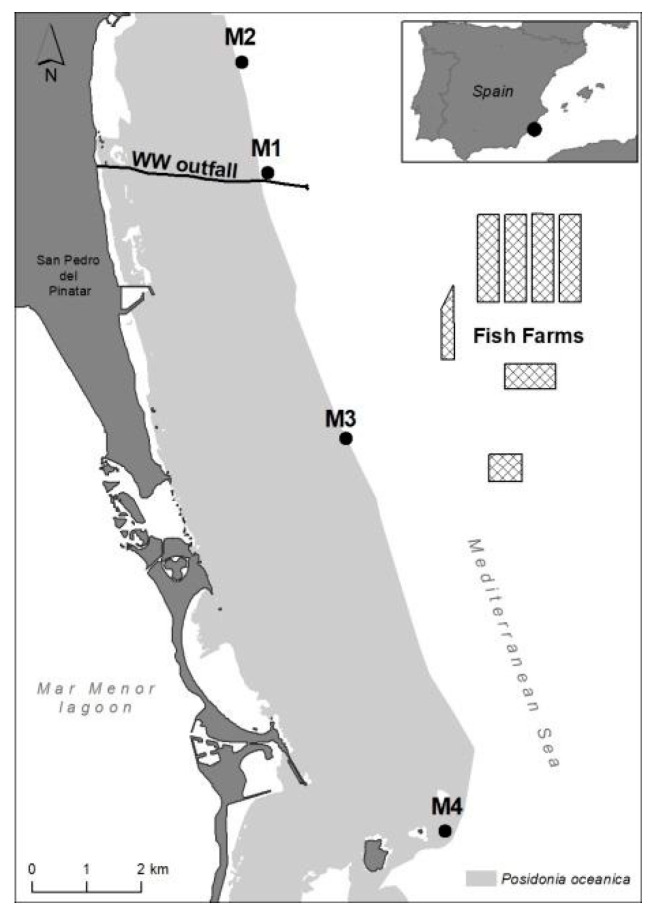
Study area showing the four sampling sites (M1–M4) along the deep margin of an extensive *P. oceanica* meadow. The seagrass distribution is shown in grey. The black line indicates the underwater outfall through which effluent wastewater treatment is discharged off-shore (WW) while the cages represent the facilities of the fish farm complex (FF).

**Table 1 plants-12-00457-t001:** Mean values of abiotic factors measured at the four studied sites (M1–M4) in summer and autumn: daily photosynthetic photon flux (DPPF, mol quanta m^−2^ d^−1^), the percentage of subsurface irradiance (% E_0_), the reduction in available irradiance relative to the reference station M4 (% DPFF_red_) and sediment organic matter content (% OM; only in autumn). Letters a–c indicate significant differences among sites (post-hoc Tukey tests, *p* < 0.05). SE: standard error.

Sites	Summer			Autumn			
	% E_0_	DPPF ± SE	% DPPF_red_	% E_0_	DPPF ± SE	% DPPF_red_	OM % ± SE
M1	10.2	4.13 ± 0.20 a	−32.8%	6.82	1.55 ± 0.13 a	−37.1%	3.49 ± 0.21 a
M2	11.66	4.43 ± 0.21 a	−28.0%	9.07	2.20 ± 0.17 b	−10.6%	3.56 ± 0.04 a
M3	11.23	4.68 ± 0.21 a	−23.9%	11.01	2.34 ± 0.15 b	−5.0%	2.11 ± 0.08 b
M4	15.33	6.15 ± 0.31 b		11.27	2.46 ± 0.14 b		1.41 ± 0.09 c

**Table 2 plants-12-00457-t002:** Summary of abiotic and biotic variations of the three *P. oceanica* meadow sites subjected to the off-shore nutrient discharges (M1, M2 and M3) relative to the reference site (M4). The grey scale indicates the intensity of the deviation, and the positive (+) and negative (–) signs indicate an increase or decrease in the variable, respectively. The white colour indicates no difference from the reference station.

Variable Type/Name	Site/Variable Response		
Abiotic factors:	**M1**	**M2**	**M3**
Daily Photosynthetic Photon Flux (DPPF)	–	–	–
% of surficial irradiance (% E_0_)	–	–	–
Sediment organic matter content (OM %)	**+**	**+**	**+**
Distance (km)	–	–	–
Biological indicators of anthropogenic influence:			
Nitrogen stable isotope δ^15^N rhizomes	**+**	**+**	**+**
%N rhizomes	**+**		
C/N rhizomes	–		
% P rhizomes	**+**	**+**	
C/P rhizomes	–	–	
Nitrogen stable isotope δ^15^N epiphytes	**+**	**+**	
% N epiphytes			
C/N epiphytes	–	–	
Ni rhizomes	**+**		
Cu rhizomes			
maximum N uptake rate (V_max_)	**+**	**+**	**+**
half saturation constant (k_m_)	**+**	**+**	**+**
Efficiency of N uptake (α _uptake)_	–	–	–
*P. oceanica* physiological variables:			
Chl *a* concentration	**+**	**+**	
Chl *b* concentration	**+**	**+**	
carotenoids concentration	**+**	**+**	
net photosynthetic rate (net-Pmax)	–	–	–
gross photosynthetic rate (gross-P)	–	–	–
respiration rate (R)	–	–	–
photosynthetic efficiency (α)		–	
saturation irradiance (Ek)	–	–	–
compensation irradiance (Ec)	–		
relative electro transport rate (r-ETR)	–		
Maximum quantum yield (Fv/Fm)	**+**	**+**	
daily metabolic carbon balance (CB)	–	–	
daily saturation period (Hk)	–	–	–
daily compensation period (Hc)			
Non-structural carbohydrates-soluble fraction	–	–	
Non-structural carbohydrates-starch fraction			
Meadow structure descriptors			
shoot density	–	–	
meadow cover	–	–	–
vertical rhizome length	–	–	
proportion of plagiotropic rhizomes	**+**	**+**	

## Data Availability

Not applicable.

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
