# Peer review of "A Critical Gap in Seagrass Protection: Impact of Anthropogenic Off-Shore Nutrient Discharges on Deep *Posidonia oceanica* Meadows"

_plants, 2023, doi:10.3390/plants12030457_

Round 1
Reviewer 1 Report
Review plants-1929439 MS
I consider the issues raised in this MS to be very relevant and important. The authors show the current deficient situation of the management of safety distances to anthropogenic sources of nutrients that are very necessary to protect highly valuable habitats such as the Posidonia oceanica meadows.
However, the MS still has some issues to overcome before considering this article ready for publication.
Major issues
1. The authors study the effects of nutrient discharges in the high seas (and want to give an estimate of the safety distance), but there are no estimates of nutrient concentration. The MS has been elaborated with biological nutrient content responses (primarily rhizome tissues), but without any data on nutrient content in water or sediment. I agree that this work supports the need to quantify the safety distance that is actually required, but they do not provide a strong case for it. Therefore, I recommend softening the argument about the causes of the observed responses.
2. The M&M and Results sections need major improvements.
Sampling design is not clear and the result section also require improvements (see the list of detailed comments).
Detailed comments
1. Subscripts, superscripts and italics are missing in the formatting of the MS.
2. Line 72. (MSFD, EC, 2012) is this a reference? Use the same format as others?
3. Lines 134-137. I dont think this paragraph is needed. It looks like the heading of table 1.
4. Line 142. ‘alpha’uptake you define this term later. Difficult to follow first time you read it.
5. Lines 151-155. The explanation of the degree of pressure influence (DPI) should be in M&M.
6. Table1. Km and alpha. These two variables increase in M1-M3 stations. It is not intuitive to think in these increases simultaneously. It would help to include a figure with the corresponding kinetics in supplementary material.
7. Line 168. “… irradiance levels in M1 and M2 were below the theoretical minimum light requirement for seagrasses…”. I don’t think is correct the comparison of 11%E0 and the curve of E as shown in figure S1. Your 11%E0 is a single value, but in reality it should be a curve as shown for M1-M4. Therefore, I am not sure if the sentence indicated is correct.
8. Figure S2. I don’t see why is this figure in supplementary material y you use it.
9. Line 188 mention changes in ammonium uptake kinetics with distance from the nutrient source, but you don’t show clearly these distances. Have you tried including distance as an abiotic variable in your analysis?
10. Line 213. To check for toxicity effects of nitrogen, I recommend to check for Fv/Fm values.
11. Lines 225, 226. First time used acronyms Hk or Hc with no previous explanation.
12. Lines 268-269. Which is the range of these variables in literature for P.oceanica from deep areas not affected by these sources of impact?
13. Figure 7. M1-M2 degraded state, M3-M4 intact integrity. Which is the threshold of intact integrity vs degraded state for your DPI. Why? How did you decide it?
14. Line 314. You work with plants from the lowest distribution limit. Of course, they are under light-limiting conditions.
15. Line 332. But it could be possible that M4 has no pigment increase due to N limitation. Isnt it? Because M2 and M3 has even more hours above Ek in fall than M4. In the discussion about light limitation in general, I think it is inefficient to use the 11%E0 argument, when you can work with real daily light dose and hours above Ek and Ec (Hk and Hc) specific for your species.
16. Lines 397-398. You don’t give any environmental value in nutrient availability. Therefore, I think you need to demonstrate there is such a nutrient excess in these stations. Otherwise, you only can suggest it might be happening.
17. Lines 407-409. You should indicate that your DPI may needs improvement to detect the loss in integrity. You have very few environmental variables included in this complex variable. I think you really need to quantify nutrient availability.
18. Line 420. “… it could have brought it into a state closer to the threshold …”. This is highly speculative. You don’t demonstrate the existence of offshore anthropogenic nutrient sources because you do not measure nutrient concentration in the environment. I agree that you provide enough information to support the need to quantify these effects, but you do not demonstrate them. For that, first, you need to provide environmental values and, second, to demonstrate that those environmental values are due to the WW or the FF sources.
MATERIAL AND METHODS (M&M)
19. Lines 472-474. Specify depth and distance to the nearest offshore discharges for every station (M1-M4).
20. Line 483. Delete ‘non-influenced’. You don't use it in the rest of the MS.
21. Lines 490 – 495. This description is insufficient. The transect design is not specified, but in the rest of biological variables you base the explanation in such transect design. What means 90m in length and 10m width? which side is in same direction to the depth slope? A figure with the sampling design would help.
22. Line 514-515. I don’t think your data are the sum of 30 days of daily doses. Something is missing or wrong.
23. Line 516. Did you measure % E0 from single measurements? It is not specified.
24. Lines 522. Sediment samples are 7 cm diameter, but how may cm deep?
25. 527-531 (and line 627). I do not understand the design of transects at all. See comment about lines 490-495.
26. Lines 532 – 537. Why nutrients are measured in rhizomes and not leaves in Posidonia? I would expect more clear patterns in leaves.
27. Lines 553 – 554. Two leaves from 6 shoots means 6 replicates per site?
28. Lines 629. rhizome height (relative to sediment level). What do you mean with relative to sediment level?
29. Lines 633-635. I do not understand the method for meadow cover.
30. Line 640. Check comment line 629. This does not match with relative to sd. Level.
31. Line 650. How did you normalize the data?
32. References: Check for mistakes (eg. Number (8) is missing the year)
Reviewer 2 Report
This is a very fine piece of work. Au can take pride in producing this research. The amount of material is immense, making life hard for the poor unpaid reviewer, but nonetheless the quality is apparent.
The contribution to management of these deep Posidonia beds is really significant, especially considering plans for even more fish farms. (When will we stop using the oceans for our toilets?!) In terms of communicating to managers...no one really believes that this work could be repeated, with the same PC results. Keep it simple, for those who will make the decisions. Pare your recs down to one or two parameters. (In another venue, I have suggested that departures of 2 per mille in 15N should trigger enforcement.)
Reviewer 3 Report
This ms is far too long with a lot of unnecessary explanations. The whole ms should be rewritten-not as a PhD thesis but as a journal paper. ITALICS SHOULD BE USED FOR ALL THE TIMES Posidonia oceanica is mentioned, e.g. lines. 16,25,41,56,67,73,79,84,4,109,110,129,etc
Line 15 Period after "habitats"
Much too many references, reduce the ten pages to about three.
Line 29"coastal shorelines"="coast".
Line 42 delete
Line 51 Period after"macroalgae"
Line 128"have been" replace with "are"
144 suggest change "Cu" to 'copper"
367 "this' change to " these"
371 "not only"... "but also" change to "and"
418 Delete 'period"
509 "underwater" should be "underwater"
533 "samples" should be "sampled"
544 "leaf"not "leave" delete "both"
545 "eaten"
558 "concentrations"
640 Should be transect"
654 "kinetic" not :kinetics"
666 New line for Conclusons"
708"who" not "which"
709 Same as 708
1100 Underwood A.J.
Round 2
Reviewer 1 Report
I think the authors have correctly answered most of the questions raised in the review and have made the corresponding corrections. Therefore, I consider this MS ready for publication with minor changes. I do not need to review again this MS.
The minor changes are related to some concerns that I still have, but I'm sure the authors can clear them up:
· The availability of light is a variable closely related to depth and to the coefficient of light attenuation (which is really what varies with anthropogenic pressure). I think it is important to include these two variables (depth and attenuation coefficient) for each work zone. In fact, I think they should specify depth variations within the 10-m transects, since depth variation may be responsible of generating noise in the biological variables, where plants are undoubtedly especially sensitive to light.
· I think there should be at least one sentence ruling out (or discussing) the effects of sea level rise in the deterioration of the distribution of this species in depth. For that it would be necessary the local SLR rates, depth of the sites and the corresponding attenuation coefficient of light in the water.
· I do not really see with the variables that have been shown that M3 is more deteriorated than M4. Less light reaches M3 than M4, but its physiological indicators are almost better than M4. That is why I believe that the DPI variable developed in this work must be interpreted very carefully. They play with variables that could be a symptom of anthropogenic pressure, but not necessarily. For example, the fact that a meadow has less light availability is not necessarily an anthropogenic effect, it may mean that it reaches deeper because the environmental conditions are better. However, in the DPI, the availability of light is being used like an anthropogenic pressure indicator. Perhaps the light attenuation coefficient would better capture the concept of human pressure (don't know).
· There are still errors in the reference list. See as examples references number 28 and 29.